# Factors of Hospital Ethical Climate among Hospital Nurses in Korea: A Systematic Review and Meta-Analysis

**DOI:** 10.3390/healthcare12030372

**Published:** 2024-02-01

**Authors:** Yoon Goo Noh, Se Young Kim

**Affiliations:** Department of Nursing, Changwon National University, Changwon 51140, Republic of Korea; ygnoh@changwon.ac.kr

**Keywords:** hospital ethical climate, environments, meta-analysis, nurse, Korean

## Abstract

In the current healthcare landscape, nurses frequently encounter various ethical dilemmas, necessitating situation-specific ethical judgments. It is crucial to thoroughly understand the factors that shape the hospital ethical climate and the elements that are influenced by this climate. This study aims to identify the variables associated with the hospital ethical climate perceived by Korean nurses. A literature search was conducted using the core database, and the effect sizes of relevant variables were analyzed using a comprehensive meta-analysis. The overall effect size analysis incorporated 56 variables, and a meta-analysis was performed on 7 variables. This study found correlations between ethical sensitivity (ESr = 0.48), moral distress (ESr = −0.30), empathy (ESr = 0.27), ethical leadership (ESr = 0.72), job satisfaction (ESr = 0.64), and intention to leave (ESr = −0.34) with the hospital ethical climate. Both personal and organizational attributes were moderately related to the hospital ethical climate. Enhancing the hospital ethical climate could positively affect both individuals and the organization. The protocol for this study has been registered with PROSPERO (CRD42022379812).

## 1. Introduction

Nurses often face complex and diverse ethical dilemmas, such as the commercialization and competition of medical care, organ donation, and life-sustaining treatment, requiring them to make ethical judgments [1]. Given nurses’ concern over ethical decisions and actions, the nursing community has developed a keen interest in the ethical climate or work environment [1]. It is known that an individual’s ethical behavior is more strongly influenced by the organizational system than by the individual knowledge of each member [2]. The ethical climate affects not only members’ behavior but also the quality of work, organization, and patient care [1,3,4]. Therefore, the importance of an ethical nursing work environment is increasingly being recognized.

The ethical climate refers to the collective understanding among organization members about what constitutes ethically appropriate behavior and how to address ethical issues. The organization’s culture and climate significantly shape the work of nurses, and they encounter an ethical climate while operating within the organization. The ethical climate aids in resolving ethical dilemmas by offering a foundation and guidance for the ethical conduct of members [5].

The ethical climate within the nursing profession is receiving increased attention, leading to a surge in research nationally and globally. The hospital ethical climate affects an individual’s moral distress [6,7], moral sensitivity [7], and moral courage [8], as well as nurses’ job satisfaction [9], their intention to remain in their roles [10], and overall organizational effectiveness [11]. Research has also been conducted to understand its effect on turnover intention [12]. The hospital ethical climate influences the culture of patient safety [13] and completeness of nursing care [14].

However, there is a recognized need to reassess this past research considering cultural diversity [1]. Research on the ethical climate of Korean nurses has primarily utilized the Ethical Climate Questionnaire (ECQ) developed by Victor and Cullen [15], consisting of nine theoretical subfactors, and Olson’s Hospital Ethical Climate Survey (HECS) [5], which comprises five subfactors. Therefore, it is evident that the item compositions of Olson’s HECS and the ECQ instruments are heterogeneous. However, comprehensive and methodologically rigorous systematic literature reviews have been conducted using the ECQ among Korean nurses [3] and the HECS in the context of international nursing [4]. Therefore, conducting a systematic review that considers Korea’s specific social and cultural aspects is necessary.

The HECS is the only tool designed to measure nurses’ perceptions of the ethical environment and is widely used in research. Furthermore, considering Korea’s cultural characteristics, comparing the influence of the ethical climate and related variables as perceived by Korean nurses could provide insights into strategies for improving the ethical climate in future research. 

This study aims to perform a systematic literature review and a meta-analysis to identify all research conducted among Korean nurses using Olson’s HECS tool that reflects Korea’s social and cultural aspects. A meta-analysis is a post-hoc analysis that integrates the findings of individual studies, providing insights into the patterns and trends in a specific research field and indicating areas where additional research is required [16]. As a result, in addition to providing a comprehensive understanding of the research on the hospital ethical climate from the perspective of Korean nurses, this study contributes to the broader understanding of the hospital ethical climate by suggesting areas for future research. Moreover, the findings of this study will serve as fundamental data for developing strategies to improve job satisfaction among nurses and retain nursing staff by understanding the unique characteristics of the hospital ethical climate as experienced by Korean nurses.

## 2. Material and Methods

This study is a systematic literature review and meta-analysis of variables related to the ethical climate of Korean nurses in hospitals. 

### 2.1. Protocol and Registration

The protocol for this study was registered in PROSPERO (https://www.crd.york.ac.uk/PROSPERO (accessed on 20 July 2023), registration number CRD42022379812). This study followed the “Systematic Literature Review Manual” by the National Evidence-based Healthcare Collaborating Agency (NECA) [17] and the Preferred Reporting Items for Systematic Reviews and Meta-Analyses (PRISMA) group’s systematic literature review reporting guidelines [18].

### 2.2. Data Sources and Searches

#### 2.2.1. Data Sources

The search source was based on the COre database recommended by the National Evidence-based Healthcare Collaborating Agency of Korea. We searched RISS, KISS, KoreaMed, KMBASE, NDSL, and KISTI for the Korean literature and MEDLINE, Cochrane, and EMBASE for the international literature [17].

#### 2.2.2. Search Strategy

The research strategy involved using specific search terms such as “Korean” AND “nurse” AND (“hospital ethical climate” OR “ethical climate” OR “ethical environment” OR “ethical work environment”) for international databases. For domestic databases, the terms used were “nurse” AND (“hospital ethical climate” OR “ethical climate” OR “ethical environment” OR “ethical work environment”). This approach ensured that the search results are relevant to the specific demographic and topic of interest. 

#### 2.2.3. Literature Selection and Exclusion Criteria

This study sought to answer, “What factors are associated with the hospital ethical climate as perceived by Korean nurses?” Using the participants, intervention, comparisons, outcomes, and study design (PICO-SD) framework, as outlined in the NECA’s manual [17], the “Participants” were Korean nurses working in hospitals, and the “Intervention” was the hospital ethical climate. There was no specific “Comparison” group, and the “Outcomes” were factors related to the hospital ethical climate. The “Study Design” included quantitative or mixed-methods research. 

The inclusion criteria for this study were as follows: the research must target Korean nurses, use Olson’s HECS to measure the ethical climate, present quantitative data related to the ethical climate (e.g., sample size, correlation coefficients, means, standard deviations, odds ratios), and be published in academic journals. The exclusion criteria were studies published in languages other than Korean or English; qualitative studies on the hospital ethical climate; and non-peer-reviewed sources, such as conference abstracts, reports, and gray literature.

### 2.3. Data Extraction

Two researchers independently conducted a literature search from September to October 2023, cross-checking their findings for consistency. They used the COre database, as recommended by the National Evidence-based Healthcare Collaborating Agency of Korea. The researchers adhered to the PRISMA flowchart for the systematic review and selection of the literature [18]. Initially, the search produced 417 papers. After eliminating 188 duplicates, they examined the titles and abstracts of the remaining 229 papers. This led to the exclusion of 204 papers (2 not written in English or Korean, 10 not related to the ethical climate, 125 not targeting Korean nurses, 45 not quantitative studies, 11 presentations at academic conferences, and 11 dissertations), leaving 25 for further review. Upon examining the full text of these 25 papers, it was identified that 19 used Olson’s HECS to measure the ethical climate. These 19 papers were then subjected to a systematic review and quality assessment, as shown in Figure 1.

### 2.4. Quality Assessment

Of the 19 selected studies, 18 cross-sectional studies were evaluated using the Joanna Briggs Institute (JBI) Checklist for Analytical Cross-Sectional Studies. One intervention study was evaluated using JBI’s Checklist for Quasi-Experimental Studies [19]. Two researchers independently carried out the quality assessment of these studies. In the event of disagreements or inconsistencies in their evaluations, the researchers engaged in a discussion to resolve these differences and reach a mutual agreement, ensuring a more reliable and objective assessment of the studies.

The assessment criteria for cross-sectional studies include clear participant inclusion guidelines, data collection time and location details, participant characteristics, exposure to disease risk factors, disease diagnosis, definition and control of confounding variables, outcome variable measurement, and the suitability of the statistical analysis methods used, totaling eight items. However, two items (exposure to disease risk factors and disease diagnosis) were deemed irrelevant to this study and thus excluded, leaving six evaluation items. Each item could be responded to with “Yes”, “No”, “Unclear”, or “Not applicable”. If a study received “Yes” for four or more items, it was deemed suitable for systematic review and meta-analysis. 

The JBI outlines the evaluation criteria for intervention studies, which include certainty of cause and effect, similarity of subjects, subject control, presence of a control group, pre- and post-intervention effect measurement, handling of treatment completion and dropout data, uniformity and reliability of effect measurement methods, and the appropriateness of statistical analysis. If a study received “Yes” for six or more of these nine items, it was considered suitable for meta-analysis.

### 2.5. Data Analysis

#### 2.5.1. General Characteristics of Papers

The systematic literature review process involved thoroughly analyzing 19 selected studies using predefined coding items. These items, which included the author, publication year, journal name, research design, conducting institution, participants, and other related factors, were independently coded by two researchers. Afterward, they swapped their coding sheets for cross-verification. If any discrepancies were found, they independently rechecked and corrected the coding. The data from this process were then analyzed using the IBM SPSS Statistics 22.0 software. The results were presented in various statistical formats: frequencies, percentages, means, and standard deviations.

#### 2.5.2. Calculation of Effect Size

This study investigated the relationship between variables related to the hospital ethical climate as perceived by Korean nurses, using a statistical measure known as the effect size, which is based on the correlation coefficient (r). The distribution of variances significantly affects the Pearson’s correlation coefficient, and as it nears one, the effect size variance decreases, potentially leading to a positive bias. To counteract this, Borenstein et al. [16] recommended using Fisher’s z transformation in the analysis instead of the correlation coefficient itself. In this study, the researchers adhered to this advice, transforming each study’s individual correlation coefficient (r) values into Fisher’s z, and then re-transforming them back into r. They also gave more weight to studies with larger sample sizes based on the assumption that effect sizes measured in larger studies are more accurate than those with smaller sample sizes [16]. The Comprehensive Meta-Analysis Software v4, Academic/Non-Profit Professional automatically handled the transformation process mentioned above (Biostat, Englewood, NJ, USA).

The correlation coefficient of the effect size may display varying correlation directions, such as negative or positive, based on the variables involved. When merging effect sizes, opposing effects could reduce the overall effect size. As such, the initial analysis in this study was performed to identify the effect sizes and their directions. When determining the overall effect size and the subfactor effect sizes, variables negatively correlated with the hospital ethical climate were reversed, and “low” was added to their variable names. Individual effect sizes were analyzed in their original direction. The significance of the calculated effect sizes was confirmed by checking whether the 95% confidence interval included zero. The effect sizes were interpreted using Cohen’s criteria (ESr ≤ 0.10: small effect; ESr = 0.3: medium effect; ESr ≥ 0.5: large effect) [20]. Homogeneity was evaluated by calculating the Q statistic and I2 value. Heterogeneity can be assumed if I2 exceeds 50% and the Q test’s *p*-value is less than 0.10 [21]. However, it has been proposed that choosing the model based on the homogeneity test results is less valid than selecting it based on whether all studies share the same effect size [16]. Therefore, considering the differences in participants, variables, and environments among the studies included in this meta-analysis, the average effect size was analyzed using a random-effects model [16]. Choosing one effect size from a single study could lead to a significant loss of information. This study mitigates this by “shifting the unit of analysis” [22]. Specifically, when calculating the overall effect size, individual studies were treated as the unit of analysis to avoid violating the independence assumption. During subgroup analyses, the effect size was treated as the unit of analysis to prevent information loss. 

#### 2.5.3. Analysis of Effect Sizes for Moderator Variables

When the analyzed studies’ effect sizes display a heterogeneous distribution, we can explore moderator variables to analyze these diverse data. To examine moderation effects, a minimum of 10 studies per moderator variable is necessary [21]. Consequently, we conducted a meta-ANOVA to analyze the effect sizes affected by these moderator variables.

#### 2.5.4. Publication Bias Analysis

Meta-analysis studies, which amalgamate individual studies, tend to publish statistically significant results more frequently [23]. To guarantee the authenticity of these results, it is necessary to analyze and authenticate any potential publication bias. This study used funnel plots and the trim-and-fill method to investigate publication bias. If the effect size, adjusted by the trim-and-fill method, demonstrated a discrepancy exceeding 10% compared to the original, this was deemed evidence of publication bias [24].

## 3. Results

### 3.1. Data Extraction

This study examined 19 papers, all published post-2013, with a sample size of 5434. These studies were primarily conducted in hospitals (16), with a few in nursing hospitals (3). The distribution of the papers over the years was as follows: one in 2013, three each in 2016 and 2017, five each in 2018 and 2019, four each in 2020 and 2021, and six up until May 2022 and 2023. Most publications (16) were from domestic journals, while the remaining 3 were from international journals. The research designs were predominantly cross-sectional studies (18), with 1 being quasi-experimental. The analysis methods varied, with regression analysis used in 14 papers, structural equation analysis in 3, odds ratio analysis in 1, and correlation analysis in 1. 

### 3.2. Characteristics of the Studies

The systematic review identified 29 related variables (Table 1). For consistency, ethical sensitivity and moral sensitivity were consolidated under the term “ethical sensitivity”. Out of the 29 variables, ethical sensitivity appeared most frequently, featuring in seven studies. This was followed by moral distress in six studies, gender in five, and religion in four. After excluding demographic characteristics such as sex, religion, marital status, and ethical education experience, the variables were sorted into two categories: personal attributes (10 variables) and organizational attributes (15).

The quality assessment of the literature yielded three cross-sectional studies with a score of 4 points, attributed to the lack of control for confounding variables. One correlation analysis study scored 5 points due to the absence of confirmed control for confounding variables, while the remaining fourteen studies scored 6 points. One intervention study scored 8 points in the “subject control” category, specifically asking, “Is the experimental group receiving a similar treatment to the control group?” As a result, all 18 cross-sectional studies and the single intervention study, making a total of 19 studies, were chosen for the meta-analysis (Table 2). 

### 3.3. Meta-Analysis

The meta-analysis incorporated all 19 papers selected via the systematic literature review. In determining the overall and subgroup effect sizes, variables with contrasting effect size directions were transformed in the opposite direction to harmonize the effect sizes. The analysis of the average effect sizes of individual variables was conducted, preserving the original positive (+) or negative (−) effect size directions.

#### 3.3.1. Overall Effect Size and Subgroup Effect Sizes

In examining the overall effect size of variables related to the hospital ethical climate among Korean nurses, 19 independent studies were individually analyzed to assess homogeneity by treating each as a separate analysis unit. These studies were found to be non-homogeneous (I2 = 96.9%, *p* < 0.001), leading to the use of a random-effects model for the analysis. The overall effect size was determined to be 0.37 (95% CI: 0.28~0.45, Z = 7.88, *p* < 0.001), which, based on Cohen’s criteria, signifies a moderate effect size (ESr ≤ 0.10: small effect; ESr = 0.3: moderate effect; ESr ≥ 0.5: large effect).

The average effect size of the demographic variables (e.g., sex, religion, marital status, and ethical education experience) was calculated using 14 homogeneous variables (I2 = 45.0%, *p* = 0.035). We analyzed these using the random-effects model and found the average effect size to be 0.11 (95% CI: 0.06~0.16, Z = 4.27, *p* < 0.001), suggesting a small effect size. In calculating the average effect size of the personal attribute variables, we included 22 non-homogeneous variables (I2 = 90.9%, *p* < 0.001). Using the random-effects model for analysis, we determined the average effect size of the personal attribute variables to be 0.40 (95% CI: 0.31~0.48, Z = 8.28, *p* < 0.001), indicating a moderate to large effect size. When calculating the average effect size of the organizational attribute variables, we included 20 non-homogeneous variables (I2 = 97.5%, *p* < 0.001). We used the random-effects model for analysis and found the average effect size of the organizational attribute variables to be 0.45 (95% CI: 0.32~0.56, Z = 6.42, *p* < 0.001), suggesting a moderate to large effect size (Table 3 and Figure 2).

#### 3.3.2. Individual Effects Sizes of Personal Attribute Variables

Variables with fewer than two studies related to them were omitted from the analysis owing to the inability to conduct a meta-analysis. Consequently, three personal attribute variables were included in the effect size calculations: ethical sensitivity, moral distress, and empathy.

Seven studies on ethical sensitivity were analyzed using a random-effects model due to their lack of homogeneity (I2 = 90.7%, *p* < 0.001). The overall effect size of ethical sensitivity was found to be 0.48 (95% CI: 0.33~0.61, Z = 5.72, *p* < 0.001), demonstrating a statistically significant and substantial positive correlation with the ethical climate in hospitals, as per Cohen’s criteria [20].

The six studies on moral distress were not homogeneous (I2 = 79.9%, *p* < 0.001) and were therefore examined using a random-effects model. The overall effect size of moral distress was −0.30 (95% CI: −0.42~−0.18, Z = −4.66, *p* < 0.001), demonstrating a statistically significant, moderate negative correlation with the ethical climate in hospitals.

Two studies on empathy were analyzed using a random-effects model owing to their lack of homogeneity (I2 = 81.9%, *p* < 0.001). The overall effect size of empathy was 0.27 (95% CI: 0.08~0.44, Z = 2.80, *p* < 0.001), which signifies a statistically significant, albeit small, positive correlation with the hospital ethical climate.

#### 3.3.3. Individual Effects Sizes of Organizational Attribute Variables 

The meta-analysis of the organizational attributes was restricted to variables with more than two associated studies. The four variables incorporated were ethical leadership, job satisfaction, intention to leave, and integrative palliative care nursing. Ethical leadership was examined in two studies, which were not homogeneous (I2 = 75.4%, *p* < 0.001) and were evaluated using a random-effects model. The overall effect size of ethical leadership was 0.72 (95% CI: 0.62~0.80, Z = 9.68, *p* < 0.001), signifying a statistically significant and substantial positive correlation with the hospital ethical climate, as per Cohen’s criteria [20].

Two studies were conducted on job satisfaction, both homogeneous (I2 = 33.0%, *p* < 0.001) and analyzed using a random-effects model. The overall effect size of job satisfaction was 0.64 (95% CI: 0.580.69, Z = 16.19, *p* < 0.001), demonstrating a statistically significant positive correlation with the ethical climate in hospitals.

Three studies focused on the intention to leave but were not homogeneous (I2 = 96.1%, *p* < 0.001) and were evaluated using a random-effects model. The overall effect size of the intention to leave was −0.34 (95% CI: −0.59~−0.04, Z = −2.24, *p* = 0.025). This suggests a statistically significant, moderate negative correlation with the hospital ethical climate.

Two studies were incorporated in assessing the impact of integrative palliative care nursing. These studies were not homogeneous (I2 = 81.2%, *p* = 0.021) and were evaluated using a random-effects model. The overall effect size of integrative palliative care nursing was 0.31 (95% CI: −0.07~0.61, Z = 1.60, *p* = 0.108), which was not statistically significant, as shown in Table 4.

#### 3.3.4. Moderator Analysis

The total effect sizes of the studies analyzed showed heterogeneity, prompting an examination of the effects of the moderator variables to comprehend the differences in effect sizes. A meta-ANOVA analysis used the type of hospital (general and long-term care or nursing facilities) as a moderator variable. A general hospital has seven or more medical departments and over 100 beds. In contrast, a long-term care hospital refers to a facility with 30 or more beds for elderly long-term hospitalized patients. This mixed-effects analysis included 15 instances of general hospitals and 3 instances of long-term care hospitals or nursing facilities. The effect size for general hospitals was ESr = 0.43 (95% CI: 0.31~0.54), and the effect size for long-term care hospitals and nursing facilities was ESr = 0.36 (95% CI: 0.03~0.59). The difference between these two groups was not statistically significant (Q = 0.36, *p* = 0.548), suggesting that the type of hospital does not moderate the relationship between the hospital ethical climate and related variables.

### 3.4. Methodological Quality

The publication bias in the studies analyzed in this study was confirmed using a funnel plot and the trim-and-fill analysis method, as shown in Figure 3. The literature used to calculate the overall effect size, the average effect size of the personal attribute variables, and the average effect size of the organizational attribute variables was included in the analysis. The distribution of the funnel plot for the overall effect size and average effect sizes of both the personal and organizational attribute variables displayed no bias. The trim-and-fill analysis method was employed to correct errors and assess the influence of publication bias on this study’s findings. This method involves removing asymmetric values from the effect size distribution, calculating a new mean effect size based on the remaining effect sizes, and filling in missing values symmetrically around the new mean effect size [24]. The black circles in the figure represent the corrected values added to the study results. If the corrected effect size changes by 10% or more compared with the uncorrected value, it is deemed that publication bias exists [24]. In this study, no corrections were needed for the overall effect size, the average effect size of the personal attribute variables, or the average effect size of the organizational attribute variables. Thus, these calculated values can be considered free from publication bias.

## 4. Discussion

This study examines factors associated with the hospital ethical climate among Korean nurses, encompassing 19 studies. The exploration of the hospital ethical climate among these nurses began in 2013, with five studies carried out between 2013 and 2018. However, from 2019 to May 2023, the number of studies consistently rose to 14, demonstrating an ongoing interest in the ethical landscape of nursing because of an increased recognition of ethical issues in nursing. Compared with the systematic review of global HECS in 2014 and 2015, when research on the hospital ethical climate among nurses was most common, Korean studies began later but have been consistently growing.

Regarding the hospital ethical climate variables among Korean nurses, 29 were identified. These consisted of 10 individual characteristics and 15 organizational characteristics, excluding 4 target characteristics. The number of organizational characteristics exceeded the individual ones. This distribution mirrors the findings of a systematic literature review of hospital ethical climates [4], which reported 21 organizational characteristics, surpassing the 11 individual characteristics.

The most common variables in the studies were ethical sensitivity, which appeared in seven studies, and moral distress, which appeared in six studies, both of which are individual characteristics. For organizational characteristics, turnover intention was found in three studies, while job satisfaction, ethical leadership, and integrative palliative care nursing appeared in two studies. An international literature review [4] found moral distress the most common variable. However, Korean nursing studies revealed ethical sensitivity as the most prevalent, followed by moral distress, suggesting a shift in focus. The same international literature review [4] reported a high turnover intention and job satisfaction (six studies each). However, these variables were less common in this study, appearing in three and two studies. The ethical climate in hospitals positively influences both individual and organizational levels. This meta-analysis and previous research [4] confirm that numerous studies have been conducted on job satisfaction and organizational commitment, highlighting the significance of hospitals’ ethical climate for organizational effectiveness. This study also included one study each on medical errors and unfinished nursing, which are potential indicators of nursing quality. Future research should consider variables related to nursing quality within the context of the hospital ethical climate.

The meta-analysis evaluated 19 studies, all deemed of moderate or high quality, thereby validating the analysis results. The variables identified about the hospital ethical climate indicate that an ethical work environment positively impacts individuals and organizations. The overall effect size of the identified variables among Korean nurses was 0.37, considered moderate according to Cohen’s criteria [20]. The mean effect size for the target characteristics was 0.11, slightly below the overall effect size. The mean effect size for the individual characteristics was 0.40, slightly above the overall effect size, and for the organizational characteristics, it was 0.45, surpassing the overall effect size. This implies that the ethical climate of a hospital has a low correlation with the target characteristics but a moderate to high correlation with the individual and organizational characteristics. Further research on the individual and organizational variables is required to confirm the positive influence of an organization’s ethical environment.

Ethical leadership among the organizational variables displayed the most significant effect size, indicating high effectiveness (ESr = 0.72). This empirically validates the strong correlation between the hospital ethical climate and ethical leadership. Acknowledging a positive hospital ethical climate led to the perception of ethical leadership in managers. The direct leadership of nurse managers significantly impacts nurses in nursing units. Guiding nursing managers according to the hospital’s ethical guidelines can reduce nurse stress, highlighting the importance of the ethical role of nurse managers [35].

Furthermore, a trusted ethical leader serves as a role model, and team members learn and emulate the leader’s behavior [36]. Thus, the ethical leadership of managers is indicative of the ethical environment in nursing. This study verified a strong correlation between managerial ethical leadership and the hospital ethical climate, highlighting the need for initiatives to foster ethical leadership among managers. Ultimately, managerial ethical leadership will enhance the hospital ethical climate.

Among the organizational variables, job satisfaction demonstrated the second-largest effect size, proving highly effective (ESr = 0.64). This offers empirical proof of a strong correlation between the hospital ethical climate and job satisfaction. The positive correlation between the hospital ethical climate and nurses’ job satisfaction [4,9,10] aligns with this study’s context. Given the strong correlation between the hospital ethical climate, which includes relationships with superiors and colleagues, and nurses’ job satisfaction, implementing interventions to enhance the hospital ethical climate could foster job satisfaction.

This study revealed that ethical sensitivity had the most significant impact among the individual variables, demonstrating a high level of effectiveness (ESr = 0.48). It also confirmed a strong correlation between the hospital ethical climate and ethical sensitivity. Nurses who exhibit ethical sensitivity are known to practice ethical nursing [37], underscoring the critical role of ethical sensitivity in this field. The hospital ethical climate provides members with a moral foundation and guidance when dealing with ethical issues within the institution [5]. The strong correlation between ethical sensitivity and the hospital ethical climate suggests that fostering a positive ethical environment within the organization can enhance ethical sensitivity and promote ethical nursing.

Conversely, turnover intention (ESr = −0.34) and moral distress (ESr = −0.30) demonstrated a moderate negative correlation, indicating a moderate association with the hospital ethical climate. The international literature review also revealed a negative correlation between moral distress [7], turnover intention [29], and the hospital ethical climate [4], which aligns with the context of this study. Given that the hospital ethical climate can exacerbate moral distress and escalate turnover intention, there is a need for research on interventions to foster a positive ethical climate.

In conclusion, this study thoroughly reviews and analyzes prior studies on factors influencing the hospital ethical climate among Korean nurses. The findings suggest that organizational variables have a larger effect size than individual variables, although the difference is not statistically significant. The individual variables with a significant, moderate, or higher effect size are ethical sensitivity and moral distress, while the organizational variables include ethical leadership, job satisfaction, and turnover intention. 

Considering the limited number of studies included in this meta-analysis and the limited number of studies examining each variable, future research should clarify the relationships between variables by conducting repeated studies that differentiate between targets and hospitals. This study offers valuable insights for managing hospital ethical climates for individuals and organizations. The implications of this study for practical application suggest efforts by organizations to improve the hospital ethical climate can enhance job satisfaction and ethical leadership and reduce the intention to leave and moral distress.

## 5. Conclusions

This study conducts a systematic review and analysis of prior research on factors influencing the hospital ethical climate among Korean nurses. Notable individual variables with a moderate or greater effect size include ethical sensitivity and moral distress, while the organizational variables encompass ethical leadership, job satisfaction, and the intention to leave the job. 

This study has several limitations. Firstly, the limited literature on several variables prevents the calculation of effect sizes for many variables. Secondly, this study is confined to those published in Korean and English, possibly omitting pertinent studies published in other languages. Furthermore, most of the studies were cross-sectional surveys, which complicates establishing causal relationships between the hospital ethical climate and associated variables. 

Based on these results, it is recommended that future research continues to examine variables associated with the hospital ethical climate among Korean nurses. Furthermore, it is suggested that subsequent studies explore the effects of interventions that foster a positive hospital ethical climate on related variables.

## Figures and Tables

**Figure 1 healthcare-12-00372-f001:**
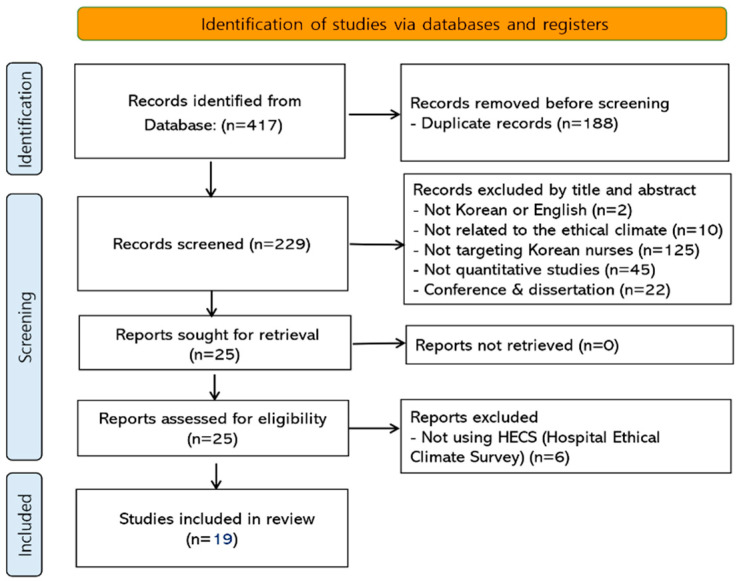
PRISMA flow diagram.

**Figure 2 healthcare-12-00372-f002:**
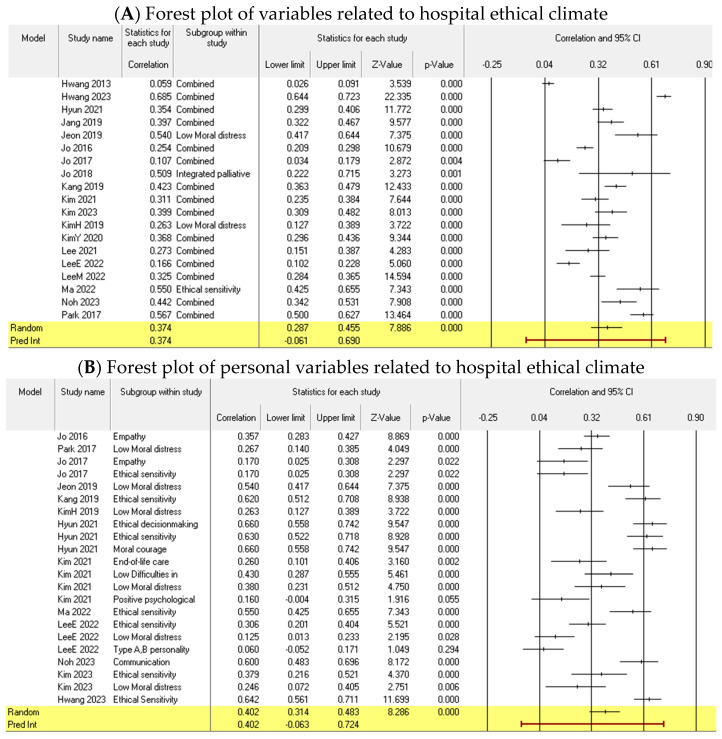
Forest plot of related factors to hospital ethical climate.

**Figure 3 healthcare-12-00372-f003:**
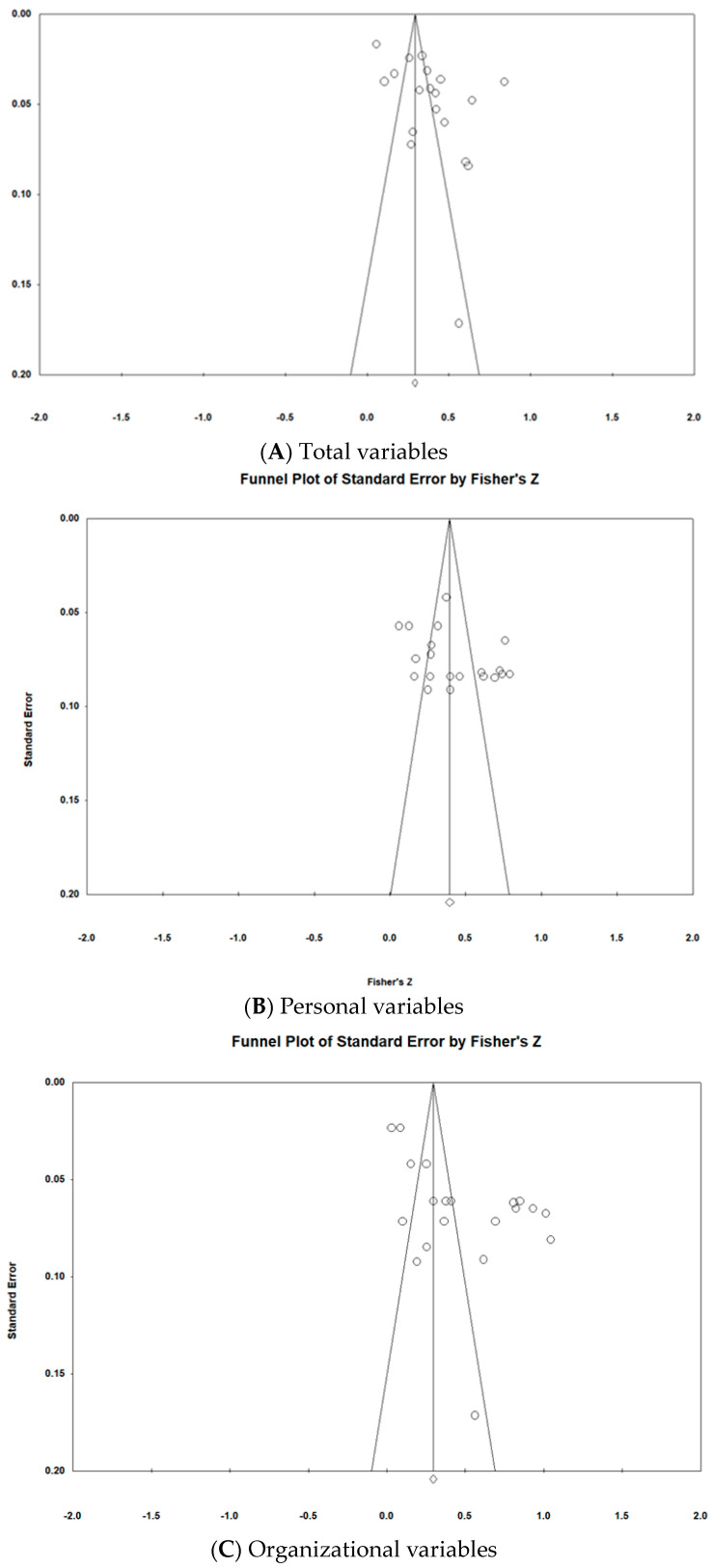
Funnel plot of selected studies for effect size extraction and imputed to adjust (standard error by Fisher’s z).

**Table 1 healthcare-12-00372-t001:** Variables related to ethical climate in selected studies.

No	Related Factors	Factor Categories	Author (Year)
1	Communication self-efficacy	P	Noh (2023) [14]
2	Difficulties in end-of-life care	P	Kim (2021) [25]
3	Empathy	P	Jo (2016) [26], Jo (2017) [27]
4	End-of-life care performance	P	Kim (2021) [25]
5	Ethical decision-making confidence	P	Hyun (2021) [8]
6	Ethical sensitivity	P	Jo (2017) [27], Kang (2019) [11], Hyun (2021) [8], Ma (2022) [28], Lee E (2022) [7], Kim (2023) [29], Hwang (2023) [30]
7	Moral courage	P	Hyun (2021) [8]
8	Moral distress	P	Park (2017) [6], Jeon (2019) [31], Kim (2019) [32], Kim (2021) [25], Lee E (2022) [7], Kim (2023) [29]
9	Positive psychological capital	P	Kim (2021) [25]
10	Type A, B personality	P	Lee E (2022) [7]
11	Comfort	O	Jo (2016) [26]
12	Emotional labor	O	Kim (2020) [10]
13	Ethical leadership	O	Park (2017) [6], Hwang (2023) [30]
14	Integrated palliative care	O	Jo (2016) [26], Jo (2018) [33]
15	Intent to leave	O	Hwang (2013) [12], Lee M (2022) [34], Kim (2023) [29]
16	Job satisfaction	O	Jang (2019) [9], Kim (2020) [10]
17	Medical error	O	Hwang (2013) [12]
18	Nursing cares left undone	O	Noh (2023) [14]
19	Nursing performance	O	Hwang (2023) [30]
20	Organizational effectiveness	O	Kang (2019) [11]
21	Organizational justice	O	Lee M (2022) [34]
22	Organizational silence	O	Lee M (2022) [34]
23	Patient safety competencies	O	Lee (2021) [13]
24	Retention intention	O	Kim (2020) [10]
25	Workplace bullying	O	Lee M (2022) [34]
26	Ethics education experience		Kang (2019) [11], Hyun (2021) [8]
27	Gender		Jo (2017) [27], Kang (2019) [11], Hyun (2021) [8], Lee (2021) [13], Lee M (2022) [34]
28	Marital status		Jang (2019) [9], Hyun (2021) [8], Lee M (2022) [34]
29	Religion		Jo (2017) [27], Kang (2019) [11], Hyun (2021) [8], Lee M (2022) [34]

O: organizational factors, P: personal factors.

**Table 2 healthcare-12-00372-t002:** Characteristics of selected studies for systematic review (*K* = 19, *n* = 5434).

No	Author (Year)	Journal Type	Design	Sample (*n*)	Setting	Related Factors	Quality Score	Factor Categories *
1	Hwang (2013) [12]	Abroad	Cross-sectional survey (logistics)	1826	Regional public hospitals	Intent to leave Medical error	6/6	O O
2	Jo (2016) [26]	Domestic	Cross-sectional survey (SEM)	567	≥500 beds hospitals	Comfort Empathy Integrated palliative care	4/6	O P O
3	Park (2017) [6]	Domestic	Cross-sectional survey	222	General hospitals and University hospitals	Ethical leadership Moral distress	6/6	O P
4	Jo (2017) [27]	Domestic	Cross-sectional survey (correlation)	182	National mental hospitals	Empathy Ethical sensitivity Gender Religion	5/6	P P
5	Jo (2018) [33]	Domestic	Quasi-experimental	37	University hospitals	Integrated palliative care	8/9	O
6	Jang (2019) [9]	Abroad	Cross-sectional survey	263	General hospitals	Job satisfaction Marital status	5/6	O
7	Jeon (2019) [31]	Domestic	Cross-sectional survey	152	University hospitals	Moral distress	6/6	P
8	Kang (2019) [11]	Domestic	Cross-sectional survey	155	150 ~< 300 general hospitals	Ethical sensitivity Organizational effectiveness Gender Religion Ethics education experience	6/6	P O
9	Kim (2019) [32]	Domestic	Cross-sectional survey	194	Long-term care hospital and facility	Moral distress	6/6	P
10	Kim (2020) [10]	Domestic	Cross-sectional survey	198	100 ~< 300 general hospitals	Emotional labor Job satisfaction Retention intention	5/6	O O O
11	Hyun (2021) [8]	Domestic	Cross-sectional survey	148	Online sampling	Ethical decision-making confidence Ethical sensitivity Moral courage Gender Religion Marital status Ethics education experience	6/6	P P P
12	Lee (2021) [13]	Domestic	Cross-sectional survey	120	Long-term care hospitals	Patient safety competencies Gender	6/6	O
13	Kim (2021) [25]	Domestic	Cross-sectional survey	144	University hospitals	Difficulties in end-of-life care End-of-life care performance Moral distress Positive psychological capital	6/6	P P P P
14	Ma (2022) [28]	Domestic	Cross-sectional survey	144	≥100 beds long-term care hospitals	Ethical sensitivity	6/6	P
15	Lee M (2022) [34]	Domestic	Cross-sectional survey	270	Hospitals	Intent to leave Organizational justice Organizational silence Workplace bullying Gender Religion Marital status	6/6	O O O O
16	Lee E (2022) [7]	Domestic	Cross-sectional survey (SEM)	308	Hospitals	Moral distress Ethical sensitivity Type A or B personality	4/6	P P P
17	Noh (2023) [14]	Domestic	Cross-sectional survey	142	<700 beds hospitals	Communication self-efficacy Nursing care left undone	6/6	P O
18	Kim (2023) [29]	Abroad	Cross-sectional survey	123	≥200 beds hospitals	Ethical sensitivity Intent to leave Moral distress	6/6	P O P
19	Hwang (2023) [30]	Domestic	Cross-sectional survey (SEM)	239	≥250 beds hospitals	Ethical leadership Ethical sensitivity Nursing performance	4/6	O P O

* O: organizational factors, P: personal factors.

**Table 3 healthcare-12-00372-t003:** Effect size of related factors on hospital ethical climate.

Categories	N or k	ESr ^†^	95% CI		Z	*p*	Heterogeneity				Analysis Model
			Lower	Upper			I2 (%)	Q	df (Q)	*p*	
Total	19 (N)	0.37	0.28	0.45	7.88	<0.001	96.9	596.11	18	<0.001	Random
Subject characteristics	14 (k)	0.11	0.06	0.16	4.27	<0.001	45.0	23.64	13	0.035	Random
Personal factors	22 (k)	0.40	0.31	0.48	8.28	<0.001	90.9	233.18	21	<0.001	Random
Organizational factors	20 (k)	0.45	0.32	0.56	6.42	<0.001	97.5	790.66	19	<0.001	Random

N = number of studies; k = number of variables; CI = confidence interval; † = correlation.

**Table 4 healthcare-12-00372-t004:** Effect size of personal factors on hospital ethical climate.

Group by	Categories	k	ESr †	95% CI		Z	*p*	Heterogeneity				Analysis Model
				Lower	Upper			I2 (%)	Q	df (Q)	*p*	
Personal	Ethical sensitivity	7	0.48	0.33	0.61	5.72	<0.001	90.7	64.77	6	<0.001	Random
	Moral distress	6	−0.30	−0.42	−0.18	−4.66	<0.001	79.9	24.97	5	<0.001	Random
	Empathy	2	0.27	0.08	0.44	2.80	<0.001	81.9	5.53	1	<0.001	Random
Organizational	Ethical leadership	2	0.72	0.62	0.80	9.68	<0.001	75.4	4.07	1	<0.001	Random
	Job satisfaction	2	0.64	0.58	0.69	16.19	<0.001	33.0	1.49	1	<0.001	Fixed effects
	Intent to leave	3	−0.34	−0.59	−0.04	−2.24	0.025	96.1	51.84	2	<0.001	Random
	Integrated palliative care	2	0.31	−0.07	0.61	1.60	0.108	81.2	5.34	1	0.021	Random

k = number of variables; CI = confidence interval; † = correlation.

## Data Availability

The data presented in this study are available on request from the first author (N.Y.G.).

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
