# Peer review of "Factors of Hospital Ethical Climate among Hospital Nurses in Korea: A Systematic Review and Meta-Analysis"

_healthcare, 2024, doi:10.3390/healthcare12030372_

Round 1

Reviewer 1 Report

Comments and Suggestions for Authors

It's not bad. Overall study design, method are appropriate although the topics have failed demonstrate the importance of it's implications or necessity. Introduction section need to revised with concise relevant statement.also Discussion section need to improve. 

Comments on the Quality of English Language

Need to check spelling and preposition,coherence, style etc.

Author Response

Comments and Suggestions for Authors

It's not bad. Overall study design, method are appropriate although the topics have failed demonstrate the importance of it's implications or necessity. Introduction section need to revised with concise relevant statement. also Discussion section need to improve.

Comments on the Quality of English Language: Need to check spelling and preposition, coherence, style etc.

  1. Introduction section need to revised with concise relevant statement.
  2. Discussion section need to improve.

Authors’ Response: We appreciate the time and effort that you and the reviewers have put into the valuable feedback and insightful comments provided on this manuscript.

We had our manuscript proofread by an American editor working at a professional editing agency, named Elsevier language editing service because writing a paper in English is always challenging to non-English speakers like us.

We have carefully considered each comment and made changes to the manuscript, as required. We have marked the revisions made to the manuscript red font.

Point 1: Introduction section need to revise with concise relevant statement.

Response 1: Thank you for your valuable feedback. In response to your comment, introduction was revised as follows.

The ethical climate within the nursing profession is receiving increased attention, leading to a surge in research both nationally and globally. The hospital ethical climate affects an individual's moral distress [6,7], moral sensitivity [7], and moral courage [8], as well as nurses' job satisfaction [9], their intention to remain in their roles [10], and overall organizational effectiveness [11]. Research has also been conducted to understand its effect on turnover intention [12]. The hospital ethical climate influences the culture of patient safety [13] and the completeness of nursing care [14]. 

However, there is a recognized need to reassess this past research considering cultural diversity [1]. Research on the ethical climate of Korean nurses has primarily utilized the Ethical Climate Questionnaire (ECQ) developed by Victor and Cullen [15], which consists of nine theoretical subfactors, and Olson's Hospital Ethical Climate Survey (HECS) [5], which comprises five subfactors. Therefore, it is evident that the item composition of Olson's HECS and the ECQ instruments are heterogeneous. But more comprehensive and methodologically rigorous systematic literature reviews have been conducted using the ECQ among Korean nurses [3] and the HECS in the context of international nursing [4]. Therefore, it is necessary to conduct a systematic review that considers Korea’s specific social and cultural aspects.

The HECS is the only tool specifically designed to measure nurses' perceptions of the ethical environment and is widely used in research. Furthermore, considering the cultural characteristics of Korea, comparing the influence of ethical climate and related variables as perceived by Korean nurses could provide insights into strategies for improving ethical climate in future research.

Point 2: Discussion section need to improve.

Response 2: Thank you for your valuable feedback. In response to your comments, the sentence in discussion was added as follows.  

The implications of this study for practical application suggest efforts by organizations to improve hospital ethical climate can enhance job satisfaction and ethical leadership and reduce intention to leave and moral distress.

Reviewer 2 Report

Comments and Suggestions for Authors

Dear authors,

I am pleased to have reviewed your article. The manuscript address important information for future researchers on the topic. The article is well written and presents the data and analysis appropriately. The research question posed is original and well defined. The data are robust and sufficient detail are provided about the methods used. The results provide a significant advance in current knowledge about the importance of ethical climate in the nursing profession.

I found this to be a very interesting paper, however, I would ask you to take into account the following suggestions for improvement.

- The methodology is aligned with the overall objective. The participant selection strategy is explained, and relevant statistical tests are used for the meta-analysis. What was the reason for using the selected databases? What was the rationale for the choice of research methodology? Please clearly define the criteria.

- In Figure PRISMA flow diagram you should change "screeiining" to 'screening'.

- The significance of the study are described in the introduction section, have you considered adding a separate sub-section for the implications of the findings for practice? Have you considered adding sub-section with the limitations and strenghts of the study?

- The references section should be reviewed to adjust it to the journal's standards.

I hope this helps

Author Response

Comments and Suggestions for Authors

Dear authors,

I am pleased to have reviewed your article. The manuscript address important information for future researchers on the topic. The article is well written and presents the data and analysis appropriately. The research question posed is original and well defined. The data are robust and sufficient detail are provided about the methods used. The results provide a significant advance in current knowledge about the importance of ethical climate in the nursing profession.

I found this to be a very interesting paper, however, I would ask you to take into account the following suggestions for improvement.

  1. The methodology is aligned with the overall objective. The participant selection strategy is explained, and relevant statistical tests are used for the meta-analysis.

What was the reason for using the selected databases?

Please clearly define the criteria.

  1. In Figure PRISMA flow diagram you should change "screeiining" to 'screening'.
  2. The significance of the study are described in the introduction section,

have you considered adding a separate sub-section for the implications of the findings for practice?

Have you considered adding sub-section with the limitations and strenghts of the study?

  1. The references section should be reviewed to adjust it to the journal's standards.

Authors’ Response: We appreciate the time and effort that you and the reviewers have put into the valuable feedback and insightful comments provided on this manuscript. We have carefully considered each comment and made changes to the manuscript, as required.

We have marked the revisions made to the manuscript red font.

Point 1: The methodology is aligned with the overall objective. The participant selection strategy is explained, and relevant statistical tests are used for the meta-analysis.

What was the reason for using the selected databases?

Please clearly define the criteria.

Response 1: Thanks for your valuable feedback. The search databases were selected according to NECA's recommendations. This information has been added to the main text as follows.

The search source was based on the COre database recommended by the National Evidence-based Healthcare Collaborating Agency of Korea. For Korean literature, we searched RISS, KISS, KoreaMed, KMBASE, NDSL, and KISTI.

Inclusion and exclusion criteria are described in the text, and the exclusion process is further described in detail.

This led to the exclusion of 204 papers (2 not written in English or Korean, 10 not relat-ed to the ethical climate, 125 not targeting Korean nurses, 45 not quantitative studies, 11 presentations at academic conferences, and 11 dissertations), leaving 25 for further review.

Point 2: In Figure PRISMA flow diagram you should change "screeiining" to 'screening'.

Response 2: Thank you for your review. The term has been corrected in Figure 1.

Point 3: The significance of the study is described in the introduction section,

have you considered adding a separate sub-section for the implications of the findings for practice?

Have you considered adding sub-section with the limitations and strengths of the study?

Response 3: Thank you for your insightful comments. We reviewed them and have added the sentence about the implication of the findings for practice in discussion rather than dividing them into sub-section as follows.

The implications of this study for practical application suggest efforts by organizations to improve hospital ethical climate can enhance job satisfaction and ethical leadership and reduce intention to leave and moral distress.

Also, we ask you for your generous understanding because the limitations and strengths of the study are presented in conclusion. 

Point 4: The references section should be reviewed to adjust it to the journal's standards.

Response 4: Thank you so much for your review. The references have been corrected.

Reviewer 3 Report

Comments and Suggestions for Authors

1. Correct the typo in Figure 1 (ex. Screeniing--> screening) 2. Also, please explain in detail the “not inclusion criteria” written in figure 1. 3. Explain the difference between hospital and nursing hospital used in data extraction. 4. Clearly present the 29 related variables in a table. 5. Please clearly indicate how many papers there are by the same author, and also describe the impact and limitations of the same author's literature.

Author Response

Comments and Suggestions for Authors

  1. Correct the typo in Figure 1 (ex. Screeniing--> screening)
  2. Also, please explain in detail the “not inclusion criteria” written in figure 1.
  3. Explain the difference between hospital and nursing hospital used in data extraction.
  4. Clearly present the 29 related variables in a table.
  5. Please clearly indicate how many papers there are by the same author, and also describe the impact and limitations of the same author's literature.

Authors’ Response: We appreciate the time and effort that you and the reviewers have put into the valuable feedback and insightful comments provided on this manuscript. We have carefully considered each comment and made changes to the manuscript, as required.

We have marked the revisions made to the manuscript red font.

  1. Correct the typo in Figure 1 (ex. Screeniing--> screening)

> Thank you for your review. The term has been corrected in Figure 1.

  1. Also, please explain in detail the “not inclusion criteria” written in figure 1.

> We have added the exclusion criteria and number of relevant papers in the text and Figure 1.

2.3. Data Extraction

~ This led to the exclusion of 204 papers (2 not written in English or Korean, 10 not relat-ed to the ethical climate, 125 not targeting Korean nurses, 45 not quantitative studies, 11 presentations at academic conferences, and 11 dissertations), leaving 25 for further review. Upon examining the full text of these 25 papers, they identified 19 that use Olson’s HECS to measure the ethical climate.

  1. Explain the difference between hospital and nursing hospital used in data extraction.

> We apologize for any confusion. We modified the term ‘nursing hospital’ to ‘long-term care hospitals’ and added the sentence for explaining the terms as follows.

A meta-ANOVA analysis was performed, using the type of hospital (general hospitals and long-term care hospitals or nursing facilities) as a moderator variable. A general hospital is defined as having seven or more medical departments and more than 100 beds, while a long-term care hospital refers to a facility with 30 or more beds for elderly ling-term hospitalized patients.

  1. Clearly present the 29 related variables in a table.

> Reflecting your comments, 29 related variables identified in 19 papers were summarized in Table 1 and added.

  1. Please clearly indicate how many papers there are by the same author, and also describe the impact and limitations of the same author's literature.

> There are some popular family names in Korea, so the same family names in authors of the selected 19 studies may appear to be the same person. As a result of checking the authors of the references, it was found that only 2 studies were published by the same person, Jo, K.H. in references 25 and 27. I ask for your generous understanding. Thank you.

Round 2

Reviewer 3 Report

Comments and Suggestions for Authors

Please clearly describe in the text the measurement tools for ethical climate measured in each study and add them to table 1. Also, please describe the variations in ethical climate resulting from these measurement tools in addition to the limitations of the study.

Please write the full name of the abbreviations in tables and figures, or present the abbreviations as footnotes. ex) HECS in figure 1

Author Response

Comments and Suggestions for Authors

  1. Please clearly describe in the text the measurement tools for ethical climate measured in each study and add them to table 1. Also, please describe the variations in ethical climate resulting from these measurement tools in addition to the limitations of the study.
  2. Please write the full name of the abbreviations in tables and figures, or present the abbreviations as footnotes. ex) HECS in figure 1

Authors’ Response: We have carefully considered each comment and made changes to the manuscript, as required. We have marked the revisions made to the manuscript red font.

Response 1: Thanks for your valuable feedback. This study analysed 19 papers using HECS, one of the tools to measure ethical climate. Therefore, I do not think there is a need to describe the tools for each paper in Table 1 or add limitations regarding differences depending on the measurement tool. We ask you for generous understanding of these matters.

Response 2: Thanks for your valuable feedback. In Table 1, the full name of the abbreviation HECS has been added.